# Scalify: scale propagation for efficient low-precision LLM training

Paul Balanca [1]   Sam Hosegood [1]   Carlo Luschi [1]   Andrew Fitzgibbon [1]

## Abstract

Low-precision formats such as float8 have been introduced in machine learning accelerated hardware to improve computational efficiency for large language models training and inference. Nevertheless, adoption by the ML community has been slowed down by the complex, and sometimes brittle, techniques required to match higher precision training accuracy. In this work, we present SCALIFY, a end-to-end scale propagation paradigm for computational graphs, generalizing and formalizing existing tensor scaling methods. Experiment results show that SCALIFY supports out-of-the-box float8 matrix multiplication and gradients representation, as well as float16 optimizer state storage. Our JAX implementation of SCALIFY is open-sourced at github.com/graphcore-research/jax-scalify.

## 1. Introduction

As the number of parameters in deep learning models have progressively increased through the years, machine learning researchers and engineers have been experimenting with the use of low precision formats for training and inference to improve computational efficiency and reduce memory usage and required bandwidth.

Starting with IEEE float16 format (the only 16-bit format supported on GPUs initially), Micikevicius et al. (2017) introduced the idea of *loss scaling* to improve stability of training and reduce the occurrence of overflow on the backward pass. This concept was later refined with the introduction of NaN and max tracking in the backward graph to dynamically adjust loss scaling depending on the training dynamics. In the landscape of low-precision techniques, loss scaling can be seen as a proto tensor-scaling method, where a global tensor scale is set on the backward gradients for numerical stability, and then compensated when the optimizer update

---

**Algorithm 1** JAX training loop using SCALIFY transform (modifications in *blue*). SCALIFY generalizes previous approaches with end-to-end scale propagation in the neural net computational graph (including optimizer).

```python
import jax_scaled_arithmetics as jsa

# Scalify transform on FWD + BWD + optimizer.
# Propagating scale in the computational graph.
@jsa.scalify
def update(state, data, labels):
    # Forward and backward pass on the NN model.
    loss, grads =
        jax.grad(model)(state, data, labels)
    # Optimizer applied on scaled state.
    state = optimizer.apply(state, grads)
    return loss, state

# Model + optimizer state.
state = (model.init(...), optimizer.init(...))
# Transform state to scaled array(s)
sc_state = jsa.as_scaled_array(state)

for (data, labels) in dataset:
    # If necessary (e.g. images), scale input data.
    data = jsa.as_scaled_array(data)
    # State update, with full scale propagation.
    sc_state = update(sc_state, data, labels)
    # Optional dynamic rescaling of state.
    sc_state = jsa.dynamic_rescale(sc_state)
```

---

is applied to the model state.

Dean et al. (2012) and Abadi et al. (2016) introduced the bfloat16 format (BF16) as a different path to improve 16-bit training stability. Compared to IEEE float16 (FP16), bfloat16 trades off mantissa bits for exponent bits, aligning with float32 exponent width (8 exponent bits compared to the 5 exponent bits of float16). In the rest of this work, we use the common notation E8M7 and E5M10 for respectively designating bfloat16 and float16 formats. Thanks to its higher dynamic range, bfloat16 has the main benefit of being a simple drop-in technique in neural net training, not requiring the complexity of dynamic loss scaling. On the other hand, as presented in Micikevicius et al. (2017); Peng et al. (2023), because of the reduction of mantissa bits, bfloat16 can not be used for master weights, normalization statistics or optimizer state, meaning that for these quantities

---

[1]Graphcore, UK. Correspondence to: Paul Balanca <paulb@graphcore.ai>.

Accepted to the Workshop on Advancing Neural Network Training at International Conference on Machine Learning (WANT@ICML 2024).

*Table 1.* Summary and comparison of most common low-precision training techniques.

| METHOD | FINE-GRAINED SCALING | SCALE PROPAGATION | NO MODEL ALTERATION |
|---|---|---|---|
| Loss scaling (Micikevicius et al., 2017) | ✗ | implicit | ✓ |
| FP8 tensor scaling (Micikevicius et al., 2022) | ✗ | ✗ | custom FP8 layers & kernels |
| Unit scaling (Blake et al., 2023) | ✓ | ✓ | additional scaling in model |
| SCALIFY | ✓ | ✓ | ✓ for FP16 training
Custom `LayerNorm` FP8 training |

most machine learning practitioners still typically default to high precision float32.

With the recent emergence of large language models (LLMs) exceeding 100 billion parameters, researchers and ML hardware vendors have explored the use of float8 formats (FP8) in matrix multiplication, further improving energy efficiency, compute time and memory footprint. The literature and practitioners (Wang et al., 2018; Sun et al., 2019; Micikevicius et al., 2022; Kuzmin et al., 2022; Noune et al., 2022) have converged towards the definition and use of two FP8 formats: E4M3 and E5M2 (i.e. respectively using 4 bits and 5 bits width exponent). The format E5M2 is used to represent the dynamic range of gradients in the backward pass, whereas E4M3 is used for activations and weights in the forward pass, that necessitate additional precision. With the introduction of these lower precision formats came the requirement of ad-hoc tensor scaling with more granularity than global loss scaling. Several papers (Noune et al., 2022; Micikevicius et al., 2022; NVIDIA, 2022) have shown that per-tensor scaling of FP8 matrix multiplication (matmul) inputs using the input signals statistics is a reliable technique that maintains the accuracy of training in higher precision. In practice, the implementation of efficient FP8 tensor scaling during training required the introduction of *delayed scaling*: tensors are scaled using the input signal statistics of the previous micro-batch, allowing scaling and statistics gathering to be done simultaneously. Finally, Peng et al. (2023) recently showed that float16 per-tensor scaling can also be used to reduce the memory footprint of master weights and optimizer state.

The main drawbacks of current approaches for FP8 training are the black-box complexity of a fused FP8 matrix multiplication and statistics gathering kernel as well as the additional compute requirement to estimate input tensor statistics at every matrix multiplication (in the forward and backward passes). The latter issue has been partially solved by the introduction of delayed scaling, but at the expense of a more complex model state management. In this work, we aim at simplifying low precision training by generalizing and automating tensor-scaling to the entire computational

graph (i.e. forward, backward and optimizer for neural net training). We introduce the SCALIFY transform which fully propagates tensor scaling information in the computational graph, bringing scaled FP8 and FP16 techniques under the same automated paradigm. Generalizing the existing approaches proposed in the literature, we believe the proposed formalization of tensor-scaling has multiple advantages:

- Systematic scale propagation decouples matrix multiplication and scaling, allowing a more efficient low-precision training schemes (reducing tensor statistics gathering);

- Integration of FP8 formats as "just another datatype": no ad-hoc black-box custom C++ kernel is required (we present in Algorithm 6 a general linear layer implementation);

- Seamless and efficient integration of scaled FP16 weights (Peng et al., 2023) and optimizer state, achieving similar training robustness and accuracy to FP32 and BF16;

- Model invariance: the SCALIFY transform does not change the semantics of the computational graph (i.e. in full precision, results are identical);

Table 1 presents a summary of the most common low-precision training methods and how they compare to SCALIFY in accuracy and ease of use.

We provide an open-source implementation of SCALIFY in JAX (Bradbury et al., 2018). In the rest of this work, we follow the JAX/XLA naming convention to describe in more details our JAX open-sourced implementation github.com/graphcore-research/jax-scalify. However, the same principles can be applied to any other ML framework (e.g. PyTorch at the ATEN operators and `torch.compile` level).

## 2. Scaled array representation and quantization

This work makes use of a "scaled array" representation, whereby a tensor $X$ is represented by a pair of arrays $(X_d, X_s)$ satisfying $X = X_d \cdot X_s$, where $X_d$ has the same shape and datatype as $X$ and $X_s$ is a tensor broadcastable to $X_d$ (in many cases, $X_s$ is a scalar). In practice, for usability by the ML practitioner, and for automated graph transformation, we represent scaled arrays using an explicit data structure `ScaledArray` as outlined in Algorithm 2. Note that there is no assumption made in Algorithm 2 on the datatype used in the array. In fact, the purpose of SCALIFY is to support out-of-the-box mixed computational graphs with scaled FP8 and FP16 tensors, generalizing the work of Peng et al. (2023).

In principle, there is also no restriction on the format used for the `scale` component in $X_s$. Nevertheless, we choose to follow the common practice of using power-of-two 8-bit scaling for the following reasons:

- Using general floating point scaling can induce additional loss of accuracy in the several rescaling operations added to the graph by the SCALIFY transform.

- Power-of-two rescaling can be implemented very efficiently in accelerated hardware, as a simple add operation on the exponent of floating point numbers.

- 8-bit power of two scaling (i.e. `E8M0` format) is the standard adopted by the Open Compute Platform (OCP, 2023) for the recent micro-scaling block formats (MXFP4, MXFP6 and MXFP8).

As the format E8M0 is not yet directly supported by modern ML accelerated hardware, in our experiments we have chosen to simulate the power-of-two scale using `float32` (the latter having 8 exponents bits, which is sufficient to represent any E8M0 value). The additional memory footprint is negligible: one float32 value per tensor.

Assuming $X$ is represented in full precision, finding the optimal `ScaledArray` low-precision representation $(q(X_d), X_s)$ of $X$ is a quantization error problem (where $q$ is the quantization method corresponding to the datatype selected). Using the signal-to-noise ratio (SNR) as the underlying metric, one wants to solve:

$$X_s = \mathrm{argmax}_{\sigma \in \mathcal{E}_8 \text{ s.t. } X = \sigma X_d} \frac{\mathbf{E}[X^2]}{\mathbf{E}[(X - \sigma q(X_d))^2]}$$
$$= \mathrm{argmax}_{\sigma \in \mathcal{E}_8 \text{ s.t. } X = \sigma X_d} \frac{\mathbf{E}[X_d^2]}{\mathbf{E}[(X_d - q(X_d))^2]},$$

where $\mathcal{E}_8$ is the set of finite (valid) values in E8M0. In other words, the quantization problem is reduced to optimal quantization of the term $X_d$, under the scale constraint $X_s \in \mathcal{E}_8$

(i.e. $X_s$ being a power-of-two). In practice, we approximate the optimal solution of this problem with a two stage algorithm:

1. Quantizing $X_d$ under the unit scaling constraint $\mathbf{E}[X_d^2] \simeq 1$, generating a floating scale tensor $X_s$;

2. Rounding $X_s$ to a power of two, and correcting accordingly the quantized tensor $q(X_d)$.

For the first step of estimating the optimal $q(X_d)$, we follow the path of the unit-scaling work (Blake et al., 2023) by using $\mathbf{E}[X_d^2] \simeq 1$ as the sweet spot for maximizing the SNR of $X_d$. In short, it falls into the optimal SNR interval for FP8 and FP16 floating point representations (see Fig. 2 in (Blake et al., 2023)), while still allowing accurate representation of large outliers (common in LLMs training and inference as shown by Dettmers et al. (2022a)). In other words, it can properly represent a tensor distribution that is the combination of a main mode (Gaussian or heavy-tailed) and large outliers. Note that Blake et al. (2023) considers the variance of tensors, but the same argument holds on the mean squared error (MSE) using the bias-variance decomposition.

Assuming `ScaledArray` inputs satisfying $\mathbf{E}[X_d^2] \simeq 1$, the SCALIFY graph transform aims at propagating this property in the computational graph, using a combination of unit-scaling rules for every operation to re-balance the $X_d$ and $X_s$ terms, and dynamic rescaling using $X_d$ tensor statistics.

In the case of neural-network (NN) training (see Algorithm 1), estimating the scale in the model state can be done as following:

- At initialization, the scale is usually known by the user (e.g. random Gaussian initialization with known variance or zeroed tensors);

- At every state update, SCALIFY will propagate a new scale into the updated state. Additionally, the scale can be dynamically re-estimated every $N$ step by gathering statistics on the current state (see `dynamic_rescale` in Algorithm 1).

In practice, as presented in Blake et al. (2023), the probability distribution of weights evolves only slowly during training. As a consequence, with a proper scale initialization of the state, our experiments presented in Section 4 did not show dynamic rescaling of the model state is required.

## 3. SCALIFY transform

The SCALIFY graph tracer propagates `ScaledArray` inputs (as defined in Algorithm 2) in the computational graph.

**Algorithm 2** `ScaledArray` data structure propagated through the graph by the SCALIFY transform

```python
@dataclass
class ScaledArray:
    """Scaled array generic representation.
        data: Main data tensor.
        scale: Scale tensor
            Usually scalar and power-of-two.
    """
    data: Array
    scale: Array

    def __array__(self) -> Array:
        """Reconstruct represented array."""
        dtype = self.data.dtype
        return self.data * self.scale.astype(dtype)
```

**Algorithm 3** Scale propagation in basic primitives

```python
def scaled_add(X: ScaledArray, Y: ScaledArray):
    # Scale using Gaussian independent assumption
    Zs = pow2_round(sqrt(X.scale**2 + Y.scale**2))
    Zd = (X.scale / Zs) * X.data
            + (Y.scale / Zs) * Y.data
    return ScaledArray(Zd, Zs)

def scaled_matmul(X: ScaledArray, Y: ScaledArray):
    # Rescaling using reduction axis size.
    rescale = pow2_round(sqrt(X.shape[1]))
    Zs = rescale * X.scale * Y.scale
    Zd = matmul(X.data, Y.data) / rescale
    return ScaledArray(Zd, Zs)
```

In the case no scaled arrays are passed (e.g. omitting the calls to `as_scaled_array` in the training loop Algorithm 1), the computational graph after SCALIFY will remain unchanged. On the other hand, passing `ScaledArray` model state and input data, the example `update` function will then return an updated `ScaledArray` model state as well as a `ScaledArray` loss (which can be trivially converted to a scalar) by propagating scaling in the forward pass, backward pass, and optimizer update.

Note that even in the case where all inputs were `ScaledArray` instances, the computational graph may still contain unscaled tensors (typically broadcasted constants), meaning that SCALIFY must be able to handle properly mixed unscaled-scaled tensors combination. Additionally, as presented in Section 3.3, supporting unscaled arrays is necessary to dynamic and delayed rescaling features.

### 3.1. **`ScaledArray` representation, bundling and unbundling**

As shown in Algorithm 2, `ScaledArray` objects are simply a pair of a main `data` array $X_d$ and a `scale` array $X_s$, broadcastable to the former. In this work, we focus on the case where the `scale` $X_s$ a scalar, but more general shapes can cover channel scaling (e.g. van Baalen et al. (2023)) and block scaling formats such as MX (OCP, 2023).

As presented above, the SCALIFY tracer should support mixed unscaled/scaled computational graphs. As a consequence, there is a need to explicitly control and represent in the graph the switch between unscaled and scaled representations of an array. For that purpose, we introduce the following two bundling/unbundling primitives to JAX:

- `set_scaling`: Always transform the input into a `ScaledArray` using the scale provided. If the input is already a `ScaledArray`, the `data` component is rescaled using the ratio of existing and new scales.

- `get_data_and_scale`: Unbundling an input tensor into its `data` and `scale` components. If the input is not a `ScaledArray` instance, the method directly returns the input paired with a constant scale 1.

Defining `set_scaling` and `get_data_and_scale` with proper no-op semantics when the SCALIFY transform is not used allows full backward compatibility in the model definition: any additional scaling logic added to improve numerical stability (e.g., any dynamic rescaling strategies presented in Section 3.3) does not change the semantic of the computational graph (i.e. the neural net model definition or optimizer update).

### 3.2. Scale propagation in basic primitives

To propagate scale arrays in the computational graph, SCALIFY requires to implement for every basic JAX primitive (or PyTorch ATEN operation) a mathematically equivalent operation on `ScaledArray`. More formally, if $f$ is a basic primitive with

$$(Y_1, \ldots, Y_m) = f(X_1, \ldots, X_n)$$

we need to define a mathematically equivalent method $f_{\text{scaled}}$:

$$(Y_{1,d}, Y_{1,s}, \ldots, Y_{m,d}, Y_{m,s})$$
$$= f_{\text{scaled}}(X_{1,d}, X_{1,s}, \ldots, X_{n,d}, X_{n,s})$$

such that if the inputs satisfy unit-scaling $\mathbf{E}[X_{i,d}^2] \simeq 1$, then the outputs also verify $\mathbf{E}[Y_{j,d}^2] \simeq 1$.

A trivial (and universal) way to satisfy the previous property would be to simply un-scale the inputs $X_1, \ldots, X_n$, and compute $Y_1, \ldots, Y_k$ by directly calling $f$, and then rescale the outputs using tensor statistics. Obviously, doing this would erase the main benefits of doing scale propagation through the graph, as the unscaling step would potentially reintroduce underflow or overflow in $X_i$ (or require higher floating point precision). Additionally, we aim at keeping

SCALIFY computationally light by default, meaning estimating $X_d$ statistics is opt-out by default (see Section 3.3 for explicit dynamic rescaling).

As a consequence, we adopt the same unit-scaling strategy as Blake et al. (2023): define a set of static rules for every primitive operation, relying only on the inputs scale and shape to approximate output scaling. To estimate the former, we employ the same simplifying assumption as Blake et al. (2023): for every operation $f$, $X_1, \ldots, X_n$ are independent Gaussian tensors. Under this hypothesis, one can estimate the scaling of the outputs $Y_1, \ldots, Y_k$ for most common NN operators. We provide in Algorithm 3 the (simplified) implementation of scale propagation for the primitives `add` and `matmul`.

As presented in Section 2 and in Algorithm 3, the scale value is always rounded to a power-of-two to avoid additional floating point errors. In our experiments, rounding down of the scale is used instead of round-to-nearest (or stochastic rounding), as experimental data tends to show it improves preservation of the unit-scaling property $\mathbf{E}[X_d^2] \simeq 1$ in the graph.

### 3.2.1. CUSTOM SCALE PROPAGATION RULE

Similarly to automatic differentiation available in ML frameworks, automatic scale propagation using basic primitive decomposition of the graph can be sub-optimal in terms of numerical accuracy and computational efficiency. As a consequence, we also provide a mechanism, `custom_scale` (similar to JAX `custom_jvp` and `custom_vjp`), to define custom scale propagation rules. The `custom_scale` feature happens to be typically useful in two common types of neural-net layers: activation and normalization layers.

In the first case, scale propagation through a complex nonlinear function like `gelu` can be brittle, whereas from the high level semantic point of view of activation functions, it is clear these functions aims at propagating positive values unchanged while clipping negative ones to zero (with a more or less smooth transition between the two regimes). While the L2 norm of the clipped tensor is smaller than the original one (meaning we could potentially assign a smaller scale to the output), we take the position of propagating the same scale to the output in order to avoid overflowing of large outliers. Appendix C.1 presents in more details the implementation of scale propagation in activation functions.

Similarly, it is inefficient and unnecessary to perform full scale propagation through a normalization layer (when excluding the affine correction part). The purpose of normalization layer is to provide an output with (approximately) mean 0 and variance 1, and as a consequence, using the scale invariance of the mean-variance normalization, it is more efficient and numerically stable to directly normal-

**Algorithm 4** Implementation of dynamic rescaling using `set_scaling` and `get_data_and_scale` primitives.

```
def rebalance_scale(arr: ScaledArray, delta: Array):
    # Apply a `delta` correction on scaling.
    _, scale = get_data_and_scale(arr)
    out_scale = scale * delta
    return set_scaling(arr, out_scale)

def dynamic_rescaling_l2(arr: ScaledArray):
    # Set scaling based on L2 statistic.
    data, _ = get_data_and_scale(arr)
    delta = pow2_round(compute_l2_norm(data))
    return rebalance_scale(arr, delta)
```

ize the component $X_d$ and assign a constant scale 1 to the output (see Appendix C.2 for more details).

Finally, the `custom_scale` feature allows properly tuning scale propagation and precision in attention layers, whether it is optimized standard softmax attention (Dao et al., 2022) or more recent alternative approaches such as Mamba (Gu & Dao, 2023). We believe that SCALIFY formalized scale propagation will incentivize ML researchers and engineers publishing new type of NN layers to investigate low precision accuracy and release optimal scale propagation rules.

### 3.3. Dynamic and delayed re-scaling with tensor statistics

A large literature on low-precision FP16 and FP8 training relies on tensor statistics gathering and dynamic rescaling to keep the training dynamic stable (see FP16 automatic-loss scaling (Micikevicius et al., 2017) and FP8 training (Micikevicius et al., 2022; Noune et al., 2022; Peng et al., 2023)). One main goal of SCALIFY scale propagation is to reduce the amount of dynamic rescaling necessary to integrate into a NN model in order to match higher precision accuracy, leading to more efficient FP8 and FP16 training schemes.

One major benefit of scale propagation is to be able to decouple dynamic rescaling of tensors from operations using FP8 low precision in the computational graph (typically matrix multiplication or reduce operations). More specifically, standard FP8 training requires dynamic rescaling of inputs at every matrix multiplication (see `TransformerEngine` implementation NVIDIA (2022)). SCALIFY scale propagation allows decoupling and finer control of when dynamic rescaling is applied, meaning the overhead of tensor statistics gathering can be reduced. For instance, in the example of Algorithm 1, we decide to update scaling at the end of the entire training iteration, following the optimizer update (with the straight-forward optimization of only doing this rescaling every $N$ micro-batches).

As presented in Algorithm 4, the implementation of dy-

namic rescaling (or delayed rescaling) can be done directly in pure Python using the two primitives `set_scaling` and `get_data_and_scale` introduced in Section 2. This highlights another benefit of SCALIFY scale propagation: the rescaling logic appears explicitly in the computation graph instead of being embedded in a low level C++ kernel (NVIDIA, 2022).

More specifically, the implementation of Algorithm 4 shows that it is useful to first introduce a generic `rebalance_scale` method operating on `ScaledArray`: instead of setting the scale, the semantic of this method is to perform a relative correction on the scale. As for `set_scaling` and `get_data_and_scale`, we note that this method is a simple no-op when applied to unscaled array, keeping backward compatibility when using rescaling inside a model definition.

Implementing dynamic rescaling using L2 moment (or any other statistic) is then a simple combination of `get_data_and_scale` and `rebalance_scale` method. Delayed rescaling can be similarly implemented by decoupling the compute of statistics and rebalancing in Algorithm 4, keeping track of the first one in the model state.

### 3.4. Handling special scaled tensors

Accurate scale propagation through an entire computational graph requires careful handling of special tensors. More specifically, we keep track in the SCALIFY tracer of two categories of tensors:

- Unscaled arrays which are broadcasted scalars;

- Scaled tensors indifferent to the scale value (i.e. combining 0, Infinite and NaN values);

Both type of tensors are commonly generated during the tracing of an ML graph (see for instance the tracing of the method `logsoftmax` in Appendix B). As they get incorporated with other scaled arrays in scaled operations, the tracer requires this additional metadata to estimate the optimal output scaling.

In the first case, broadcasted scalars are commonly generated by the JAX (or PyTorch) graph tracer from scalar constants (with potential intermediate unary operations applied). When combined with scaled array inputs, the SCALIFY tracer needs to be able to estimate a scale (or otherwise will throw an error). In the general case, this would require gathering expensive statistics, but in the case of a broadcasted scalar, it is trivial: it simply corresponds to a split between the mantissa and the exponent in the floating point representation.

In the second case, scaled tensors invariant to the scale value,

*Table 2.* GPT2 model and training configuration

| MODEL | |
| --- | --- |
| # of parameters | 168M |
| dimension | 1024 |
| $n$ heads | 16 |
| $n$ layers | 8 |
| TRAINING | |
| dataset | WikiText-103 |
| max learning rate | $2 \cdot 10^{-4}$ |
| learning rate schedule | cosine |
| optimizer | Adam ($\beta_1 = 0.9$, $\beta_2 = 0.95$) |
| batch size | 32768 |
| $n$ tokens | 800M ( 6 epochs) |
| warmup | 1 epoch |

these tensors are typically appearing in masking and error correcting parts of the computational graph (i.e. combine with a `select` operation). Keeping track of which tensors are invariant to scale allows more accurate scale propagation as the SCALIFY tracer knows it can just use the scale value from other inputs to properly estimate the output scaling. In practice, we reserve a special encoding ANY_SCALE in $\mathcal{E}_8$ to propagate this information.

## 4. Experiments and results

In this section, we assess the effectiveness of SCALIFY approach for training GPT-style LLMs. Our experiments focus on the training of a GPT2 (Radford et al., 2019)-like model with 168M parameters on the WikiText-103 dataset (Merity et al., 2017). The configuration of the decoder-only Transformer (Vaswani et al., 2017) architecture and training hyper-parameters are detailed in Table 2. Our JAX implementation is heavily inspired by open-source NanoGPT repositories (Karpathy, 2023; Garcia, 2023).

We aim to address three main questions with our GPT2 training experiments:

- Is SCALIFY a drop-in replacement of loss scaling strategies for pure FP16 training?

- Does FP8 SCALIFY training accuracy match higher precision, with minimal dynamic rescaling introduced?

- Can master weights and optimizer state be stored in scaled FP16 using SCALIFY?

We summarize the configurations of our main experiments in Table 3, and present in Figure 1 the profile of training losses. In all experiments, scale propagation is performed end-to-end using SCALIFY, following the simplified training loop presented in Algorithm 1.

*Table 3.* SCALIFY FP8 and FP16 experiments, decoupling low precision use in compute, master state, gradients and optimizer state. Except for the FP32 baseline, `ScaledArray` are used in state representation and propagated end-to-end in the computational graph.

| EXPERIMENT | MATMUL (GEMM) | MASTER STATE | STATE GRADS. | OPTIMIZER STATE | DYNAMIC RESCALING | TRAINING LOSS |
|---|---|---|---|---|---|---|
| FP32 baseline #0 | FP32 | FP32 | FP32 | FP32 | x | $2.87 \pm 0.12$ |
| SCALIFY FP16 #1 | FP16 | FP32 | FP16 | FP32 | x | $2.87 \pm 0.12$ |
| SCALIFY FP16 #2 | FP16 | FP16 | FP16 | FP32 | `LayerNorm` bwd | $2.92 \pm 0.12$ |
| SCALIFY FP8 #3 | FP8 | FP16 | FP8 | FP32 | `LayerNorm` bwd | $2.92 \pm 0.12$ |
| SCALIFY FP8 #4 | FP8 | FP16 | FP8 | FP16 | `LayerNorm` bwd & grads | $2.93 \pm 0.12$ |

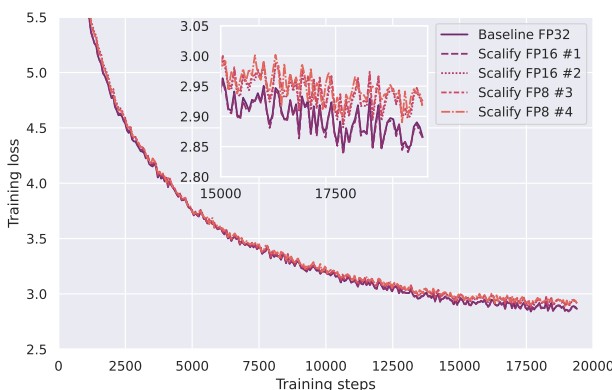

*Figure 1.* Training loss of SCALIFY GPT2 experiments. Table 3 details the low-precision settings used in these experiments.

Our first experiment SCALIFY FP16 #1 shows that SCALIFY can be used as drop-in replacement of loss-scaling for FP16 training. In this setting, the main advantage of SCALIFY is to avoid the addition of another hyper-parameter (loss scaling), or a complex automatic loss scaling strategy (i.e. no dynamic rescaling of activations, gradients or state is required). We believe that even this most simple SCALIFY setting could be of interest to machine learning practioners, as a drop-in replacement of BF16 while benefitting from additional precision brought additional mantissa bits in the IEEE FP16 format.

For our experiments #2, #3 and #4, we added dynamic rescaling on the backward pass of `LayerNorm` layers in every transformer block. The main motivation and justification is the following: `LayerNorm` layers are implicitly dynamically re-scaling activations on the forward pass (see Appendix C.2, ensuring that `ScaledArray` inputs to the multi-head-attention and feed-forward residual paths are well calibrated. As a consequence, FP8 matmuls on the forward pass have already accurately scaled inputs. On the other hand, no operation in the computational graph is providing (implicit) dynamic rescaling of gradients, meaning

that despite the use of unit-scaling type rules described in Section 3.2, the distribution of the gradients will tend to shift away from the ideal unit-scaling property $\mathbf{E}[X_d^2] \simeq 1$ in the backward pass graph. As a consequence, it is natural to insert some dynamic rescaling operations on gradients in the backward pass, and we decided to follow a similar strategy to forward pass normalization by adding one in each residual path of the Transformer layer. Algorithm 5 presents the simple implementation of the custom `LayerNorm` layer with dynamic rescaling of input gradient.

With the additional `LayerNorm` gradient dynamic rescaling, experiment SCALIFY FP16 #2 shows similarly to the work of Peng et al. (2023) that per-tensor scaling FP16 can be used for master weight representation without loss in accuracy, optimizing memory footprint during training.

Experiments SCALIFY FP8 #3 and #4 aim to demonstrate that SCALIFY supports as well out-of-the-box low precision FP8 training. In the first one, we use FP8 to speed-up matrix multiplications and reduce memory footprint of the weight gradients. Similarly to the previous FP16 training experiment, matching higher precision training only requires 2 dynamic rescaling in every `Transformer` layer instead of 18 (all forward and backward matrix multiplications) in common FP8 training approaches such as NVIDIA (2022).

Finally, as presented in Peng et al. (2023), we show that reducing the optimizer state precision to FP16 can also be achieved with SCALIFY, at the cost of adding additional dynamic rescaling of the state gradients. This is in line with the findings of the latter work which also presents a dynamic rescaling strategy of FP8 gradients, called "auto-scale", in order to match higher precision accuracy. We believe this last experiments highlights one of the main benefit of SCALIFY in terms of usability and expressibility for ML practitioners: it only requires a 2 lines code change (cast of the optimizer state and 'dynamic_rescale' of gradients) to integrate low precision optimizer state representation in a training loop.

In our experiments, dynamic rescaling of the model state

**Algorithm 5** Custom LayerNorm implementation with backward gradient dynamic rescaling.

```python
class CustomLayerNorm(flax.linen.LayerNorm):
    """Custom LayerNorm with backward
    gradient dynamic rescaling.
    """
    @compact
    def __call__(self, x):
        # Dynamic rescaling of input gradient
        # No-op from the model def. perspective.
        x = jsa.ops.dynamic_rescale_l2_grad(x)
        # FLAX default LayerNorm implementation.
        return super()(x)
```

such as presented Algorithm 1 was not necessary to match the baseline accuracy. It still needs to be investigated whether larger scale LLMs training would require it.

## 5. Related work

Most large models are trained with mixed precision (Micikevicius et al., 2017; NVIDIA, 2019) where half precision is used for most operations but sensitive operations such as gradient accumulation and softmax operations are maintained in FP32. Generally loss scaling is required to ensure gradients remain stable. Stability issues have also been noted when using FP16 for training large models (Zhang et al., 2022; Brown et al., 2020). Therefore, bfloat16 is often preferred for its increased dynamic range (Kalamkar et al., 2019; Almazrouei et al., 2023) and block scaled data formats have been developed to increase dynamic range while maintaining low precision (Rouhani et al., 2023). It is generally required to keep the optimizer state in high precision. However, there has been work reducing this to 8 bits by employing blockwise quantization methods (Dettmers et al., 2022b). Some recent methods have gone further to reduce mixed precision down to FP8 for many operations (Peng et al., 2023; NVIDIA, 2022), however often complex heuristics are required to determine when low precision can be used or overhead is introduced by empirically rescaling tensors to maintain stability. The work most closely related to ours is Unit Sclaing (Blake et al., 2023) which makes similar assumptions about how scale propagates. Unit scaling uses these assumptions to change the model, whereas SCALIFY maintains separate scaling factors alongside a consistently-scaled representation, leaving the model definition unchanged.

Many methods have been developed to quantize LLMs for interference (Bondarenko et al., 2021; Frantar et al., 2023). These methods require varying degrees of finetuning or knowledge distillation to achieve similar performance to their high precision baselines and there is usually a tradeoff

between degree of quantization and inference performance. Quantization to 8 bits (Xiao et al., 2024) can be achieved with little drop in accuracy and several methods have been developed to quantize more aggressively to 4 bits (Lin et al., 2024; Zhao et al., 2024) or beyond (Kim et al., 2024). Although these extremely low precision methods generally lead to accuracy loss compared to FP16 baselines (Wu et al., 2023), they allow extremely efficient deployment of large models. Efficient hardware support is generally only available for quantization to powers of 2. Nonetheless, work has been done to alleviate this issue by developing kernel design schemes with provide support for arbitrary quantization bit widths (Xia et al., 2024).

## 6. Conclusion

In this work, we explore end-to-end scale propagation in neural net training. We introduce SCALIFY, a transform automatically performing scale propagation in a computational graph, formalizing and generalizing several previous works on low precision FP8 and FP16 training. Our experiments demonstrate that using unit-scaling type scale propagation, SCALIFY allows out-of-the box training FP8 LLM training, with minimal tensor dynamic rescaling required. Additionally, it supports FP16 scaled tensor representation of model and optimizer state, improving memory usage in large models training. SCALIFY integrates seamlessly in JAX ML framework without black-box custom operations and kernels, allowing machine learning practitioners to easily customize their low-precision training setup. In future work, we plan to scale up the size and training of the LLM models and extend to more recent architectures such as Llama.

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

**Algorithm 6** General neural net Linear layer, with its FP8 specialization

```
def general_linear_layer(
        x: Array, w: Array, bias: Array, *,
        fwd_dtype: DType, bwd_dtype: DType):
    # Forward casting. No-op on gradient.
    w = cast_on_forward(w, fwd_dtype)
    x = cast_on_forward(x, fwd_dtype)
    # Matrix multiplication, with
    # potentially different output dtype.
    out = jnp.dot(x, w)
    # Backward casting. No-op on activation.
    out = cast_on_backward(out, bwd_dtype)
    # Adding bias, using output precision.
    out = out + bias
    return out

# Using E4M3 for weights + activations.
# Using E5M2 for backward gradient.
linear_layer_fp8 = partial(general_linear_layer,
    fwd_dtype=ml_dtypes.float8_e4m3fn,
    bwd_dtype=ml_dtypes.float8_e5m2fn)
```

## A. General linear layer

Independently of per tensor scaling, the recent introduction of FP8 formats E4M3 and E5M2 requires a more general definition of the Linear layer used in neural net models. More specifically, FP8 matrix multiplications differ in two ways from classic 16-bit and 32-bit matrix multiplications:

- Mixed inputs: FP8 hardware supports matrix multiplication between mixed E4M3 and E5M2 inputs, on the contrary to 16-bit matmuls (i.e. no mixed FP16/BF16);

- Higher precision output: accumulation is performed in higher precision than FP8 in machine earning hardware, meaning FP8 matmuls output by default higher precision than FP8 (FP16 or BF16 typically).

This more general setting is a consequence of the machine learning research (Noune et al., 2022; Peng et al., 2023) on low-precision training which has demonstrated that neural net activations and backward gradients have different statistical distribution, and thus, require different FP8 formats (E4M3 and E5M2 respectively).

As a consequence, we present in Algorithm 6 a more general definition of a Linear layer, supporting any kind of compute format. As presented in the code, it only requires a more granular parametrization of the data types used in the forward activation and weight, and the backward gradient, which can be done using casting operators cast_on_forward and cast_on_backward that act respectively only the forward or backward passes.

As mentioned above, Algorithm 6 general Linear layer

**Algorithm 7** `logsumexp` decomposition in JAX primitives.

```
{ lambda ; a:f32[3,5]. let
    b:f32[3] = reduce_max[axes=(1,)] a
    c:bool[3] = is_finite b
    d:f32[3] = broadcast_in_dim[
        broadcast_dimensions=() shape=(3,)] 0.0
    e:f32[3] = select_n c d b
    f:f32[3] = stop_gradient e
    g:f32[3,1] = broadcast_in_dim[
        broadcast_dimensions=(0,) shape=(3, 1)] f
    h:f32[3,5] = sub a g
    i:f32[3,5] = exp h
    j:f32[3] = reduce_sum[axes=(1,)] i
    k:f32[3] = abs j
    l:f32[3] = log k
    m:f32[3] = add l f
  in (m,) }
```

definition is completely independent of tensor scaling, on the contrary to existing FP8 training approaches such as NVIDIA (2022) which requires combining the two aspects in the same complex low level C++ kernel. We believe this decoupling and simplification of FP8 layers and APIs will help ML practitioners adopt more widely low precision training.

## B. `logsumexp` JAX decomposition

`logsumexp` is an example of a function requiring proper handling a special values tensors (see Section 3.4). More specifically, as shown in Algorithm 7, it presents the typical case of a broadcasted 0 constant then used in a `select` operation to correct for potential invalid values. In order to handle properly scale propagation in the call `e:f32[3] = select c d b`, SCALIFY tracer needs to know that the tensor `d` is actually a scalar tensor filled with 0, meaning that any scale is valid for the former and the `select` primitive can just directly propagate the second operand `b` scale.

## C. Custom scale propagation in some common neural net layers

We detail in this section the implementation of custom scale propagation in activation and normalization layers.

### C.1. Activation layers

As mentioned in Section 3.2.1, it is more efficient and accurate to use custom scale propagation for activation layers instead of automatic SCALIFY graph tracing (similarly to custom backward pass instead of automatic differentiation for activation functions).

Common activation functions in deep learning (i.e. relu,

gelu, swish, ...) can be represented as:

$$f(X) = X \cdot g(X)$$

where $g$ is a (bounded) gating function satisfying $\lim_{-\infty} g = 0$ and $\lim_{+\infty} g = 1$.

As presented in Section 3.2.1, it is reasonable to propagate the same scale through an activation function. The main question remaining is how to implement the scaled version efficiently and accurately. From the decomposition above, one can simply write:

$$f_{\text{scaled}}(X_d, X_s) = (X_d \cdot g(X_d \cdot X_s), X_s).$$

Compared to the unscaled version $f$, $f_{\text{scaled}}$ only requires an additional (scalar) multiplication $X_d \cdot X_s$ in low precision. Additionally, in the case the previous product overflows $\pm\infty$, the estimate will still be accurate as we know that $g$ is well defined on $\pm\infty$ values, meaning that conversion to a higher precision format such as FP32 is not required.

Scaled backward gradient propagation can be estimated similarly using:

$$f'(X) = g(X) + X \cdot g'(X) \quad \text{where } \lim_{\pm\infty} X \cdot g'(X) = 0.$$

$f'$ is therefore a bounded function, meaning that based on the chain rule, we can similarly propagated the backward gradient scale unchanged as a first order approximation.

### C.2. Normalization

Common normalization layers are the composition of a mean-variance normalization followed by an affine transform. We approximate scale propagation in the former as following:

$$\frac{X - \mathbf{E}[X]}{\sqrt{\mathbf{Var}[X] + \varepsilon}} = \frac{X_d - \mathbf{E}[X_d]}{\sqrt{\mathbf{Var}[X_d] + \frac{\varepsilon}{X_s}}}$$

$$\simeq \frac{X_d - \mathbf{E}[X_d]}{\sqrt{\mathbf{Var}[X_d] + \varepsilon}}.$$

Based on the last line estimate, we therefore implement scale propagation through a normalization layer by normalizing its `data` component, and assigning scale 1 to the output.

From our perspective, the approximation introduced in the previous equation is actually a more accurate estimate of the normalization of a tensor in the case of very small variance. More specifically, assuming the simple case where $X$ is a sampled from a normal distribution $\mathcal{N}(m, \sigma)$, and the scale is estimated accurately (i.e. $X_s \simeq \sigma$), then our estimate will return as expected a tensor approximately sampled from $\mathcal{N}(0, 1)$ for any $\sigma > 0$, whereas the original normalization will converge to zero as $\sigma \to 0$ due to the numerical stability term $\varepsilon$.

