# OpenReview forum: "Scalify: scale propagation for efficient low-precision LLM training"
_ICML.cc/2024/Workshop/WANT — WANT@ICML 2024 Poster_

### Official Review · Reviewer_mP1C · 2024-06-10

**Confidence:** 4

**Summary:**

The paper presents a useful tool for using-point quantization, where tensors are represented using a value tensor and a shared scales vector.

It covers technical details for the underlying implementation of fundamental operations, and explains how this tool can be used in practice to simplify the workflow with FP8 models.

**Strengths:**

* The work is useful: it addresses a genuine missing component for FP8 quantization in software. Several works have used FP8 with scaling, but most have re-implemented it themselves without clear standards. I can see myself using this tool in the future.

* Detailed: It is very common for FP8 works to suggest scaling as an afterthought, without discussing the implications on computation. This work is helpful at delving into the hidden details of the scaling operations, which has more complexity than others would suggest.

**Weaknesses:**

* As a paper describing a practical tool, the work is not, by itself, novel.

* The experimental section lacks enough details to put it in context. For example, I am not sure whether the experiment is using scalar, channel or MX scaling, all were reported to work well when the tool was explained, in previous sections.

**Suggestions:**

* From my experience (and some works), E3M4 can surpass M4E3 when properly scaled, especially in transformer based networks. I wouldn't brush it aside.

---

### Official Review · Reviewer_FXPW · 2024-06-12

**Confidence:** 4

**Summary:**

The paper presents a principled way of unifying different low precision training methods under a single paradigm which they call Scalify. It is currently enabled with Jax, with the authors mentioning a similar potential for enabling this in PyTorch via aten ops. The authors present principled methods to enable scaling of tensors in low-precision regimes for both 16-bit and 8-bit training. For 8-bit training, they build on top of unit scaling from Blake et al, and also show ways to unify with the recently proposed OCP formats and more recent work on FP8 training from Peng et al. Another aspect the authors point out is the simplification from writing explicit kernels to integrating the method into the framework itself.

Based on the coherency of the paper and relevance to training in lower-precisions for scaling to larger models, I'd recommend an accept.

**Strengths:**

1. The paper presents the Scalify approach, which is straightforward and easy to integrate with existing libraries such as Jax / PyTorch.
2. The authors provide thorough explanations for the different components for training with the Scalify approach and also mention how existing training workflows can be integrated with Scalify (algo. 1) to enable training with different precisions.
3. The use of unit scaling as a baseline to improve on top of is good, as it provides a more principled approach to integrating their scaling methodology.
4. The paper clearly states when to enable further customizations for normalizations when using lower precision, providing better efficiency + stability for training.
5. The paper also provides primitives for low precision addition, which is useful in the case of biases and residuals (if needed).
6. Finally, they show combining dynamic per-tensor scaling through the use of two functions make it easier to handle and how the scaling can be optional in the main training loop (last line Algo 1). An interesting experiment though the authors may have considered is to show an example of a run with / without dynamic tensor scaling to show the strengths of their approach.

**Weaknesses:**

1. While this is primarily a system + framework paper with the focus on unification of low-precision methods, here are some things I wish the authors presented better:

a. There are very few results. I understand that LLMs results can be bottlenecked on compute, but it would have been good to understand how the proposed methods scale to at least 1 order of magnitude higher model (~1B params) and if their proposed approach would scale or not.
b. The results for low-precision training (FP8 #3 / #4) see some minor degradation on loss (though within std. deviation of fp32 baseline), seeing some potential ICL evaluation would have given a better sense of how savings through memory + efficiency are translating to downstream tasks.
c. Given that their methodology provides a systematic way for scaling without statistics etc. it would have been good to understand even for their existing training runs, memory requirements for storing scalars across all arrays and any potential upside through speedup because of cycles not spent on statistics gathering etc.

**Limitations:**

One main limitation is that implementing Scalify requires deep knowledge at the framework level (for example, aten ops are not something the everyday practitioner is working with when using pytorch for example). While there is no incentive for the authors, they might consider releasing their code, allowing for at least users of Jax to take advantage of the features the library provides.

---

### Official Review · Reviewer_enGd · 2024-06-14
**Review for Scalify: scale propagation for efficient low-precision LLM training**

**Confidence:** 1

**Summary:**

This paper introduces Scalify, an end-to-end paradigm for scale propagation in computational graphs, which generalizes and formalizes current tensor scaling techniques. Experimental results demonstrate that Scalify enables seamless use of float8 for matrix multiplication and gradient representation, as well as float16 for storing optimizer states.

**Strengths:**

The algorithms are described clearly and the experiment results support the conclusion.

**Weaknesses:**

Limitation of the method is not discussed much.

---

### Meta-Review · Area_Chair_Gwi2 · 2024-06-15

**Recommendation:** Accept (Poster)
**Confidence:** 4

**Metareview:**

I recommend acceptance.

To encourage more researchers to use Scalify, I encourage Authors:
1. Discuss and show more results in the future, especially the large model runs.
2. Discuss precision w.r.t Attention mechanisms (standard Softmax Attention, and other new attentions, e.g. Linear Attention, Mamba etc, as these linear attentions are more sensitive with precision issues)
3. Discuss if Scalify can handle training/evaluation in case of state sharding (especially with tensor parallel) without too much extra effort.

---

### Decision · Program_Chairs · 2024-06-17

**Decision:**

Accept (Poster)

**Comment:**

We thank the authors for their time and contribution to WANT and we are pleased to share that after the reviewing process the paper has been accepted. Congratulations! We encourage the authors to consider reviewers' feedback for the improvement of the camera-ready version. We hope to see you in person at the workshop and brainstorm on efficient training research together!